# Coproducing healthcare service improvement for people with common mental health disorders including psychotic experiences: a study protocol of a multiperspective qualitative study

Alexandros Georgiadis,[1] Robbie Duschinsky,[2] Jesus Perez,[3] Peter B Jones,[4] Debra Russo,[4] Clare Knight,[4] Emma Soneson,[4] Mary Dixon-Woods[1]

[1]THIS Institute (The Healthcare Improvement Studies Institute), University of Cambridge, Cambridge, UK
[2]Applied Social Science Group, University of Cambridge, Cambridge, UK
[3]CAMEO Early Intervention Services, Cambridgeshire andPeterborough NHS FoundationTrust, Cambridge, UK
[4]Department of Psychiatry, University of Cambridge, Cambridge, UK

**Correspondence to**
Dr Alexandros Georgiadis; ag2000@medschl.cam.ac.uk

## ABSTRACT

**Introduction** Some people, who have common mental health disorders such as depression and anxiety, also have some psychotic experiences. These individuals may experience a treatment gap: their symptoms neither reach the increasingly high threshold for secondary care, nor do they receive full benefit from current interventions offered by the Improving Access to Psychological Therapies (IAPT) programme. The result may be poorer clinical and functional outcomes. A new talking therapy could potentially benefit this group. Informed by principles of coproduction, this study will seek the views of service users and staff to inform the design and development of such a therapy.

**Methods and analysis** Semistructured interviews will be conducted with IAPT service users, therapists and managers based in three different geographical areas in England. Our sample will include (1) approximately 15 service users who will be receiving therapy or will have completed therapy at the time of recruitment, (2) approximately 15 service users who initiated treatment but withdrew, (3) approximately 15 therapists each with at least 4-month experience in a step-3 IAPT setting and (4) three IAPT managers. Data analysis will be based on the constant comparative method.

**Ethics and dissemination** The study has been approved by the London Harrow Research Ethics Committee (reference: 18/LO/0642), and all National Health Service Trusts have granted permissions to conduct the study. Findings will be published in peer-reviewed academic journals, and presented at academic conferences. We will also produce a 'digest' summary of the findings, which will be accessible, visual and freely available.

## INTRODUCTION

Some people, who have common mental health disorders such as depression and anxiety, also have some psychotic experiences, such as attenuated paranoia or voice hallucinations.[1–7] These experiences are likely indicators of severity of mental distress and associated trauma. For

---

### Strengths and limitations of this study

► This study will address the needs of a currently underserved group: people with a common mental disorder who also have psychotic experiences.
► The multiperspective approach will provide insights into limitations of current treatments and inform the development of a new, more targeted talking therapy.
► The study will not include real-time observations of therapeutic encounters which might generate more insight into how service users and therapists interact and coconstruct care.

---

example, people with psychotic experiences are at least seven times less likely to reach remission of depressive symptoms, are at increased risk of self-harm[1–5] and show poorer response to standard psychological treatments for depression and anxiety, even in combination with pharmacotherapy.[8–10] Despite the clinical relevance of psychotic experiences in anxiety and depression, and the evidence that psychosis, depression and anxiety may share common aetiologies,[11–18] current psychiatric diagnostic classifications do not adequately acknowledge the presence of psychotic experiences in common mental disorders. This has important consequences, not least because psychiatric diagnostic classification systems are used to configure how mental health services are organised in England, which in turn impacts on how those services are accessed and used.

The Improving Access to Psychological Therapies (IAPT) programme in England is particularly important for addressing common mental disorders. IAPT provides steps 2 and 3 in the National Institute for

**BMJ**

Health and Care Excellence four-step approach to the treatment of common mental disorders.[19] Cognitive–behavioural therapy (CBT) is the predominant, but not the only approach adopted by these services, particularly for depression where there is a wider range of recommended treatments (counselling, couples therapy, interpersonal therapy, brief psychodynamic therapy and CBT).[20] At step 2, psychological well-being practitioners provide referral advice and CBT treatments to people with persistent subthreshold symptoms and mild to moderate common mental disorders. At step 3, high intensity therapists treat people with moderate to severe common mental health disorders and those who do not show sufficient improvement in response to initial interventions at step 2.

All IAPT services routinely collect patient-reported outcomes measures. The 9-item Patient Health Questionnaire (PHQ-9)[21] and 7-item Generalised Anxiety Disorder (GAD-7)[22] measures for depression and anxiety, respectively, used at baseline and following each treatment session allows the calculation of 'reliable recovery'. This is judged to occur if a person scores below the clinical threshold on PHQ-9 (score ≤10) and GAD-7 (score ≤8) post-treatment, and also shows a significant improvement in their condition.[23] Psychotic experiences are not currently routinely measured in IAPT services, although a recent study has shown that approximately 30% of people treated in IAPT services had psychotic experiences.[24] Reliable recovery remains at around 48% of the IAPT population as a whole,[23] but those with a common mental disorder and psychotic experiences may be much less likely to recover,[24] again suggesting greater severity of illness.

People with a common mental disorder including psychotic experiences thus may face a substantial gap in services; their symptoms do not reach the increasingly high threshold for secondary care[25] but the standard CBTs used in IAPT do not target psychotic experiences nor the trauma histories with which they are often associated.[8–10] Focusing on mood disturbance alone, as IAPT services currently do, is likely to result in psychotic experiences being left untreated, and could contribute to poorer clinical and functional outcomes.

Just as the condition itself (common mental disorder with psychotic experiences) is scattered across different chapters of Diagnostic and Statistical Manual of Mental Disorders (DSM-5),[26] evidence-based psychological treatments are scattered between different service settings that are inadequate to assess or treat the condition in a holistic manner. By reconfiguring and assembling such treatments into a new talking therapy for use in the low stigma setting of IAPT services, the prospects of recovery might be greatly improved. To maximise effectiveness and applicability within the IAPT setting, the new talking therapy should be grounded in a thorough understanding of the priorities and views of IAPT service users, therapists and managers, and should be based on principles of coproduction.

## Coproduction: a collaborative approach to healthcare improvement

Based on relationships of reciprocity and mutuality among citizens, service users and clinicians, coproduction seeks to improve health and support the development of user-led, people-centred health services. Viewing users as active agents and employing an asset-based model, in which narratives and experiences of illness, services and treatment contribute to health research, service improvement and knowledge, it is firmly rooted in the principle that bringing together clinical, lay and biomedical knowledge allows new forms of values, knowledge and social relations to emerge. Coproduction frameworks are increasingly used as a means to guide the reform and improvement of mental health services,[27–32] with the National Health Service (NHS) England Five Year Forward View for Mental Health specifying that coproduction between managers, clinicians and service users should be at the heart of service design. For example, Memon *et al*[33] interviewed 26 adults from black and minority ethnic backgrounds to examine perceived barriers to accessing mental health services, with the aim of informing the development of effective services to improve equity in healthcare. Similarly, Bee *et al*[34] explored the views of professionals on how to improve service users and carer involvement in mental healthcare planning. Accordingly, the design and execution of our investigation will be based on the involvement of key stakeholders to facilitate the identification and in-depth examination of key priorities for improvement.

## The study

The proposed study is part of a wider research programme grant led by the Department of Psychiatry, University of Cambridge. As part of this wider research programme, the proposed study is expected to inform the design and development of a new talking therapy that aims to improve the prospects of recovery for people with a common mental disorder including psychotic experiences who are attending IAPT services.

## Aims and objectives

The study is located within the interpretivist tradition of scientific enquiry exploring individuals' subjective experiences and meaning-making processes of social reality.[35] The focus of qualitative methodologies on the meaning and meaning-making processes of social phenomena and events[36] is consistent with the aims of our proposed study, which are to:

1. Produce an in-depth account of both the challenges and positive experiences reported by IAPT users who have a common mental disorder that includes psychotic experiences in receiving care from current step-3 IAPT services, including their views on their therapist(s), the talking therapy they were offered, how it was implemented, how far they saw it as effective in addressing their mental health needs, and their ideas, values and priorities for the development of a

**Table 1** Inclusion and exclusion criteria for people who are in therapy or have completed their therapy

| Inclusion | Exclusion |
|---|---|
| Male or female<br>18 years of age or over<br>PHQ-9 score ≥10 and/or GAD-7 score ≥8 (at any point during the therapy)<br>CAPE-P15 scores ≥1.47 for frequency and ≥1.47 for distress<br>Expressed an interest in participating in research either via the CAPE-P15 or IAPT registration form<br>Have completed or attended at least four therapy sessions at the time of recruitment | PHQ-9 score ≤10 and/or GAD-7 score ≤8<br>CAPE-P15 scores ≤1.47 for frequency and ≤1.47 for distress<br>Have not expressed an interest in participating in research neither in CAPE-P15 not IAPT registration form<br>Have attended fewer than four therapy sessions at the time of recruitment |

CAPE-P15, Community Assessment of Psychic Experiences-Positive 15 items; GAD-7, 7-item Generalised Anxiety Disorder; IAPT, Improving Access to Psychological Therapies; PHQ-9, 9-item Patient Health Questionnaire.

new talking treatment and for improving therapists' skills in treating service users with a common mental disorder including psychotic experiences.

2. Produce an in-depth account of both the challenges and positive experiences of IAPT therapists in delivering care for IAPT users with a common mental disorder including psychotic experiences, including therapists' understanding of the term 'common mental disorder including psychotic experiences', and their views on the value, role and effectiveness of the talking therapy they currently use and of clinical skills in addressing such manifestations, as well as their ideas, values and priorities for the development of a new talking treatment and for improving therapists' skills in treating service users with such experiences.

3. Produce an in-depth account of IAPT managers' experiences in delivering and supporting the delivery of care for IAPT users with a common mental disorder including psychotic experiences, including managers' views on the current pathways of care and their effectiveness, their impact on meeting performance and recovery targets, as well as managers' ideas, values and priorities for the development of a new talking treatment and for improving therapists' skills in treating service users with common mental disorders including psychotic experiences.

## METHODS AND ANALYSIS
### Study design
The study will conduct semistructured interviews to explore the views of IAPT service users and staff.

### Recruitment, sampling and selection of service users for interview
The proposed study will take place in three IAPT provider sites based in the East and Southeast of England. We aim to recruit up to approximately 30 service users who meet our criteria for a common mental disorder including psychotic experiences from across the participating IAPT provider sites (see tables 1 and 2). Around half (n=15) will be in therapy (at least four sessions) or will have completed therapy at the time of recruitment; the other half will have initiated treatment but withdrawn. We believe that this number will be sufficient to achieve theoretical saturation,[37] whereby no further themes emerge as additional participants are included, but we will monitor this carefully. We will collect demographic data of service users, such as age, gender and ethnicity. We will use quota sampling to ensure diversity of gender and ethnicity, by matching the proportion of subgroups to the population (eg, roughly half of the sample will be female participants). We will then choose potential participants by adhering to the subgroup population proportion.

**Table 2** Inclusion and exclusion criteria for people who initiated treatment but withdrawn

| Inclusion | Exclusion |
|---|---|
| Male or female<br>18 years of age or over<br>PHQ-9 score ≥10 and/or GAD-7 score ≥8 (at any point during the therapy)<br>CAPE-P15 scores ≥1.47 for frequency and ≥1.47 for distress<br>Expressed an interest in participating in research either via the CAPE-P15 or IAPT registration form<br>Initiated treatment but withdrawn | PHQ-9 score ≤10 and/or GAD-7 score ≤8<br>CAPE-P15 scores ≤1.47 for frequency and ≤1.47 for distress<br>Have not expressed an interest in participating in research neither in CAPE-P15 nor IAPT registration form |

CAPE-P15, Community Assessment of Psychic Experiences-Positive 15 items; GAD-7, 7-item Generalised Anxiety Disorder; IAPT, Improving Access to Psychological Therapies; PHQ-9, 9-item Patient Health Questionnaire.

## Sampling

The research team has liaised and worked collaboratively with IAPT managers and therapists from the participating IAPT provider sites to agree on the most convenient procedures for service user recruitment, which will minimise the effort required from IAPT staff while maximising the likelihood of reaching the above recruitment targets. The proposed study will focus on adult female and male service users, as people below 18 years old have different mental healthcare structures and service delivery within IAPT and are not the focus of our study. Non-English speakers will be included.

We will employ a two-step method to help us ensure that we identify, approach and recruit the right sample while reducing the risk of causing unnecessary distress to service users. First, in addition to assessment measures currently used for depression (PHQ-9) and anxiety (GAD-7), the participating IAPT provider sites administer the shortened version of the Community Assessment of Psychic Experiences questionnaire Positive-15 items (CAPE-P15)[38] as part of a service evaluation to identify the prevalence and impact of psychotic experiences in their current caseloads. All service users attending step-3 IAPT services are offered the opportunity to complete the self-reported CAPE-P15 questionnaire once during the course of their treatment.

The CAPE-P15 is a 15-item, self-report measure of experiences that are similar to positive psychotic symptoms, such as perceptual abnormalities, persecutory ideation and bizarre experiences. It measures the frequency (ranging from 1-never to 4-nearly always) and level of distress (ranging from 1-not distressed to 4-very distressed) that each psychotic experience causes to the individual. To account for non-response to any items, scores are weighted for the number of valid answers. The weighted score is the sum score divided by the amount of items completed (ie, the mean of completed items). Higher scores indicate a higher frequency of psychotic experiences and an increased level of distress in relation to these experiences. A cut-off point ≥1.47 for frequency and a cut-off point of ≥1.47 for distress has been useful for detecting people who have psychotic experiences.[39] Service users who have scored in ≥1.47 for both frequency and distress in CAPE-P15 will be considered as cases, and therefore, eligible to participate in our study.

Second, at the end of the CAPE-P15, we have added a query where we ask service users to indicate, by ticking a box, whether they are interested in taking part in the proposed study and discuss their experiences of using step-3 IAPT services. From among those service users with scores of ≥1.47 for both frequency and distress on CAPE-P15, we will approach a proportion of those service users (following the quota sampling method) who indicated in the CAPE-P15 questionnaire that they would like to participate in the proposed study. We will also include service users who have indicated in their IAPT registration form that they want to take part in research projects. To ensure confidentiality, we will use service users' IAPT numbers (unique identification numbers, similar to NHS numbers, but exclusively used within IAPT services). This approach will allow us to approach the right sample and minimise the risk of unduly stressing potential participants, as well as ensure that service users' data and confidential information are protected.

## Data collection

Semistructured interviews will be conducted using a topic guide that is coproduced with our service user advisory group, and that is informed by a review of relevant literature. Members of our service user advisory group and members of the research team developed the topic guide, and in so doing sought to minimise the risk of unduly stressing participants through content and delivery. The topic guide aims to elicit participants' views on the identification and treatment of people with a common mental disorder including psychotic experiences attending step-3 IAPT services. More specifically, it asks participants to discuss relevant aspects of their personal and life stories, including how past negative life events contributed to their mental health difficulties, and their positive and negative experiences of receiving care from IAPT services. The researcher will explore service users' views on what improvements might be made at the levels of the therapist and talking treatment. The interview will also examine how such improvements should be implemented and prioritised to improve the care that step-3 IAPT services deliver to people with a common mental disorder including psychotic experiences.

## Recruitment, sampling and selection of IAPT therapists and managers for interview

Our aim is to recruit five IAPT therapists and one manager per IAPT provider. We will present details of the study during team meetings and ask for volunteers, with more than 6 months experience working with step-3 IAPT service users. We will collect data on therapists/managers' years of professional practice, gender, ethnicity and clinical orientation. We will use a purposive sampling method to ensure diversity of therapists and managers participants.

## Data collection

Semistructured interviews will be conducted using a topic guide which has been coproduced with an IAPT manager and also informed by a review of the relevant literature. The manager and members of the research team developed the topic guide collaboratively, with the aim to develop interview questions and prompts that covered the full scope of topics pertinent to the scope of the study. The topic guide aims to elicit participants' views on the identification and treatment of people with a common mental disorder including psychotic experiences attending step-3 IAPT services. More specifically, the topic guide will ask participants to discuss their positive and negative experiences of delivering care to service users with a common mental disorder including psychotic

experiences, the talking therapies they use to treat and care for these users, and their reflections regarding effectiveness. The researcher will also explore the types of knowledge that participants draw on to provide care and support to these service users. Additionally, the interview will ask about therapists' and managers' ideas, values and priorities for the development of a new talking treatment and for improving therapists' skills in treating service users with a common mental disorder including psychotic experiences.

### Data analysis

All interview data will be coded by one researcher experienced in qualitative research methods. Although we will not formally test inter-rater reliability, a sample of the collected interview data (n=5) will be double coded by another researcher. Analysis will be based on the constant comparative method as described by Charmaz *et al*.[37] We will draw on the sensitising constructs identified through the literature, but we will remain open to the emergence of other important themes which may be more relevant to the primary care mental health setting. NVivo software (NVivo V.11, QSR International, Doncaster, Australia) will be used to facilitate data coding. The combination of the different strands of data (ie, semistructured interviews with IAPT users, therapists and managers) will be brought together to inform the development of a new talking therapy for people with a common mental disorder including psychotic experiences. More specifically, our qualitative study findings will inform the development of the talking therapy prior to its piloting in preparation for a randomised controlled trial of the therapy. We will also use the findings to ensure that participants' priorities are incorporated in the theory of change of the intervention.

### Limitations of the study

The study has several limitations. First, the expected findings may not be applicable to people younger than 18 years old. Second, the sample size will be small, so appropriate caution will be needed in the interpretation and generalisation of the findings. Third, the study will not include real-time observations of therapeutic encounters—which might generate more insight into how IAPT service users and therapists interact and coconstruct care (ie, access to enacted, not just espoused, valuations and preferences). Fourth, the recruitment and sampling methods may influence the expected findings, as service users who meet the inclusion criteria may not respond to our invitation due to the severity of their mental health problems or they may not be selected to participate due to the limited capacity of the researcher to interview all participants who meet the inclusion criteria.

### Patient and public involvement

The McPin Foundation (http://mcpin.org) convened a service user advisory group for the programme. This involves nine people with common mental disorders including psychotic experiences from diverse backgrounds, including ethnic minorities. For this particular work package, the research team work collaboratively with the service user advisory group to develop the IAPT service user invitation letter, information sheet and interview topic-guide list. All study participants will receive a digest summary of findings written in lay English language.

## ETHICS AND DISSEMINATION

The following permissions have been obtained for this study:
1. NHS Permissions were granted by all Trusts involved.
2. We registered the study on the UK Clinical Research Network Study Portfolio (reference: 38180).

### Ethical considerations

A number of steps will be taken to ensure that the research is conducted ethically, protecting participants' rights and maintaining appropriate confidentiality of the information provided.

### Voluntary nature of participation and right to withdraw

Information sheets for participants, as well as researcher verbal reassurance, will clarify that participation in the study is voluntary. No pressure will be placed on any individual to take part in the research. It will be made clear to participants that their decision to take part or decline to take part in any part of the research will affect neither the care they receive nor their legal rights. Participants will also be informed about their right to withdraw from the study.

### Informed consent

All participants will be asked to give their informed consent before any data are collected. Such consent will only be sought after they have been provided with full information about the research and what their participation would involve. Additionally, they will be given sufficient time to consider the aims and scope of the study and ask questions. Service users will not be approached about the research if—in the opinion of mental health professionals—(1) they are unable to give informed consent, (2) their current mental health gives significant cause for concern and (3) their mental health may be adversely affected by taking part in the research. The researcher will liaise with the participating IAPT teams to ensure that the potential participants selected to be contacted and invited to participate in an interview do not meet any of the above criteria.

### Safety of participants and researcher

Consideration has been given to ways in which taking part in the research might be harmful to participants (in particular service users) and steps will be taken to manage such potential outcomes should they occur. Clinical risk assessments are routinely undertaken within IAPT services to identify during the initial assessment anyone at high risk of harm to themselves or to other people. High-risk

individuals will be referred on to appropriate services and, accordingly, not be included within our sample. Nevertheless, reflecting on mental health, traumatic experiences and events, and past healthcare experiences may cause distress and measures will be taken to ensure the safety of participants. Should participant distress occur during the interviews the researcher will pause the interview, and support the participant to manage the distress and regulate their emotions. The researcher will then check if she/he is still willing to continue with the interview process. If a participant remains distressed, the researcher will ask for permission to contact the participant's general practitioner (GP) or mental health worker, while, encouraging the participant to seek further support that may be available in their environment. The researcher will have up-to-date contact information for psychiatric emergency services, if required. This will include access to First Response Services and/or Crisis Resolution Home Treatment Teams. These services should be able to advise about how to manage difficult clinical scenarios.

If, during the interview, a participant suggests that they intend to harm themselves or another person, the researcher will ask for the contact details of the participant's GP and inform their GP or mental health worker. The researcher will inform the participant about this action and the reasons for doing this. As stated above, the researcher might also contact psychiatric emergency services in the area. This limit to confidentiality will be explicit in the research information sheet, and it will be explained again before the start of the interview. We believe that the risk of psychological distress to the IAPT therapists and managers will be minimal.

Consideration has also been given to the safety of the researcher who will carry out the interviews. The researcher has attended training to equip him with the skills, awareness and knowledge required for the safe conduct of research in community settings. He is also experienced in carrying out research with vulnerable individuals, including offenders, homeless people and people with severe mental illness. The researcher will follow The Healthcare Improvement Studies Institute (www.thisinstitute.cam.ac.uk) lone worker policy when interviewing people at their homes.

## Dissemination

We will report the findings to national and international, multidisciplinary audiences to inform future research and practice using a variety of methods. We will disseminate original articles in peer-reviewed academic journals, conference presentations and learning reports. We will also produce a 'digest' summary of the findings, which will be accessible, visual and freely available.

## STUDY STATUS

Data collection started in June 2018 and is expected to last 7 months.

## DISCUSSION

Improving the prospects of recovery for all people experiencing mental health disorders is a priority in the mental health policy agenda in England. Within this context, this study aims to inform the development of a new talking therapy that will increase the prospects of recovery for people with a common mental disorder including psychotic experiences.

**Acknowledgements** The authors would like to thank the McPin Foundation, the TYPPEX Lived Experience Advisory Panel and the participating research sites for helping us to set up this research.

**Contributors** Conception or design of the work: PBJ, JP, RD and MD-W. Drafting the article: AG, RD and MD-W. Critical revision of the article: AG, RD, JP, PBJ, DR, CK, ES and MD-W. Final approval of the version to be published: AG, RD, JP, PBJ, DR, CK, ES and MD-W.

**Funding** This work was supported by the National Institute for Health Research grant number RP-PG-0616-20003. This work was also supported by Mary Dixon-Wood's Wellcome Trust Investigator award WT09789 and by THIS Institute, which is supported by the Health Foundation. Mary Dixon-Woods and Peter B Jones (emeritus) are National Institute for Health Research (NIHR) senior investigators. This report is independent research funded by the National Institute for Health Research (Programme Grants for Applied Research).

**Disclaimer** The views expressed in this publication are those of the author(s) and not necessarily those of the NHS, the National Institute for Health Research or the Department of Health.

**Competing interests** None declared.

**Patient consent** None required.

**Ethics approval** NHS Ethics – approval granted by London Harrow Research Ethics Committee (reference: 18/LO/0642).

**Provenance and peer review** Not commissioned; externally peer reviewed.

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
