## [Reviewer comments · BMJ Open]

ARTICLE DETAILS

TITLE (PROVISIONAL)	Co-producing healthcare service improvement for people with common mental health disorders including psychotic experiences: a study protocol of a multi-perspective qualitative study.
AUTHORS	Georgiadis, Alexandros; Duschinsky, Robbie; Perez, Jesus; Jones, Peter; Russo, Debra; Knight, Clare; Soneson, Emma; Dixon-Woods, Mary

VERSION 1 – REVIEW

REVIEWER	Sue Patterson Metro North Hospital and Health Service, Australia and Griffith University, Australia
REVIEW RETURNED	22-Aug-2018

GENERAL COMMENTS	A well written protocol, clearly expressed, study addressing an important area using appropriate methodology... only suggestions that may be considered by authors are to describe epistemological position and commentary on limitations of the approach and methods. I would also appreciate a brief account of how the data/analysis will inform the development of the new talking therapy. There are also minor typographical errors. I look forward to reading results of the study.
--

REVIEWER	Angelo Barbato Istituto di Ricerche Farmacologiche Mario Negri-IRCCS, Milano, Italy
REVIEW RETURNED	05-Sep-2018

GENERAL COMMENTS	This is a well written interesting paper. I agree with you that in UK NHS there is a treatment gap for people with common mental disorders experiencing also psychotic symptoms. Your qualitative study may provide insights on the views of both users and providers on appropriate service provision for this underserved group. You correctly write that narratives of illness experiences and providers' views may contribute to service quality improvement within a co-production framework. However, you fail to provide examples from recent research on qualitative studies used to guide change in mental health services. The only reference on co-production in your list is on cystic fibrosis (Sabadosa et al., 2014). A brief discussion of studies in mental health area, with updated references, would improve your paper. Moreover, you should give more information on the data you will collect to describe the population of users and providers participating in your study, including the clinical and theoretical orientations of therapists.
--

VERSION 1 – AUTHOR RESPONSE

Responses to reviewers' comments	
Reviewer 1	Our response
1. A well written protocol, clearly expressed study addressing an important area using appropriate methodology	Thank you.
2. State epistemological position	We have included a statement about the epistemological position of our study. Page 6, first paragraph under the heading 'Aims and Objectives'
3. Provide a commentary on limitations of the approach and methods	We have added a sub-section where we discuss the limitations of our study. You will find this sub-section in page 12 (last paragraph).
4. Provide a brief account of how the data will inform the development of the new talking therapy	We have added a brief account of how the data will inform the development of the new talking therapy. You will find this information in page 12, towards the end of the sub-heading 'Data analysis'
5. Address typographical errors	All typographical errors have been addressed.
Reviewer 2	
1. This is a well written interesting paper	Thank you.
2. Provide examples from recent research on qualitative studies used to guide change in mental health services	We have included some examples of recent mental health related research and updated the reference list. You will find these changes in page 5 (at the middle of the paragraph) under the sub-heading 'Co-production: a collaborative approach to healthcare improvement'.
3. Include more references in the co-production list (mental health related)	We have added more mental health related references about co-production. You will find these changes in page 5 (at the middle of the paragraph) under the sub-heading 'Co-production: a collaborative approach to healthcare improvement'.
4. Give more information on the data we will collect to describe the population of users and providers participating in the study	We have included more information on the data we will collect for service users (see end of page 7, beginning of page 8) and therapists/managers (see top of page 11).